# Urban Birds Using Insects on Front Panels of Cars

**Jukka Jokimäki *** and **Marja-Liisa Kaisanlahti-Jokimäki**

Arctic Centre, University of Lapland, P.O. Box 122, 96101 Rovaniemi, Finland
* Correspondence: jukka.jokimaki@ulapland.fi

**Simple Summary:** Urbanization changes the living conditions and resources of birds in many ways. One important resource is the availability and quality of food. Due to air pollution, there might be a lack of insect food for birds in cities. Our main aim was to study which species used this food source, how widespread it is and when it started. Our study indicates that seven urban species, especially sparrows and corvids (crows, jackdaws and magpies), use insects smashed on the front panels of cars in Finland. This behavior was detected for the first time during the year 1971 in Finland, about 40 years later than the first global observation made in London. In general, this behavior was concentrated on the urban parking places of hypermarkets with many cars and on late breeding phases of birds. It is possible that birds compensate for the lack of high-quality insect food in cities by using the insect material provided by the cars from the surrounding countryside.

**Abstract:** Urbanization influences the food availability and quality for birds in many ways. Although a great amount of food for birds is provided incidentally or intentionally in urban areas, the quantity of insect-based food can be reduced in cities. We studied the role of one artificial food source, insects smashed on the front panels of cars, in Finland, and more specifically in the city of Rovaniemi, by conducting questionnaire research, searching for data from databases and performing a field study. Our results indicated that a total of seven bird species have been detected using insects on the front panels of cars in Finland. However, this behavior is not yet common since about 60% of responders to the questionnaire stated that this behavior is currently either rare or very rare. Most of the observations identified House Sparrows, followed by the White Wagtail or the Eurasian Jackdaw. Only a few observations identified the Eurasian Tree Sparrow, the Hooded Crow, the Great Tit and the Eurasian Magpie. The phenomenon was distributed quite widely across Finland, except in the case of the Eurasian Jackdaws, for which observations were restricted only to the southern part of the country. The first observation was made about the House Sparrow in 1971, followed by the White Wagtail (1975), Hooded Crow (1997), Eurasian Jackdaw (2006), Eurasian Tree Sparrow (2011), Eurasian Magpie (2019) and Great Tit (2022). The species using this food source are mainly sedentary urban exploiters, such as corvids and sparrows, that have been previously reported to have several different types of innovative behaviors. Most of the observations were conducted in urban parking sites of hypermarkets, and no observations were made in residential areas. Most of the foraging observations were made during the end phase of the breeding season, partly supporting the extra need for high-quality insect-based food for nestlings and fledglings. Our observations indicate that this behavior is not yet common and widespread among species.

**Keywords:** feeding innovations; cities; food; foraging; sparrows; crows; parking areas



## 1. Introduction

Urbanization changes the structure and composition of habitats and causes many kinds of disturbances (such as pollution, noise and artificial light [1]) that consequently influence predictability, abundance and quality of food sources for birds [2,3]. For some species, such as omnivores and species using feeding sites, artificial food availability can be positive factor [4,5], whereas for some species, such as insectivores, may suffer from

decreased food availability [6]. In general, urban exploiter birds have been reported to have a broader dietary niche than urban avoiders [7] and correspondingly, species with a high degree of trait specialization are likely have a lower capacity to persist in urban ecosystems [8]. During the last few decades, a decrease in insect numbers has been reported e.g., in Europe [9]. This decrease has been explained by the intensification of agriculture and pesticide use [9]. In urban areas, air pollution is an important factor in decreasing the insect abundance [9]. Despite the fact that urban bird species are generally flexible in their food choices and are quite independent from their food preferences [7], nestlings of most species need high-quality protein-rich food to secure their good growth and development [10]. There are several examples that have reported negative influences of the junk food on the nesting success and development of young birds [2,11]. Therefore, it is very important that urban birds find high-quality food, at least during their breeding season [12–14].

By increasing the green cover and water surface, urban planners and managers can partly support insect availability and abundance for insectivores [15–18]. However, in most urban areas, grey areas (buildings, roads and parking areas) cover a greater proportion of city area than either green, blue (water) or brown (brownfields) areas [1]. In addition, due to the poorer air quality, loss of suitable host plants, intensive insecticide use, habitat loss and fragmentation, the insect abundance in urban areas can seldom be as high as in more natural areas ([18–20]; but see opposite results in [21]). However, cars and trains might incidentally transport insects from the more natural areas of the urban surroundings to the highly urbanized city areas. For example, a great number of insects are killed on car windscreens [22,23]. When people move from more natural or rural areas to cities with their cars, for example for shopping, insects massed on windscreen, radiator grilles or number plates offer protein-rich food for those species and individuals that are able to use them (Figure 1).

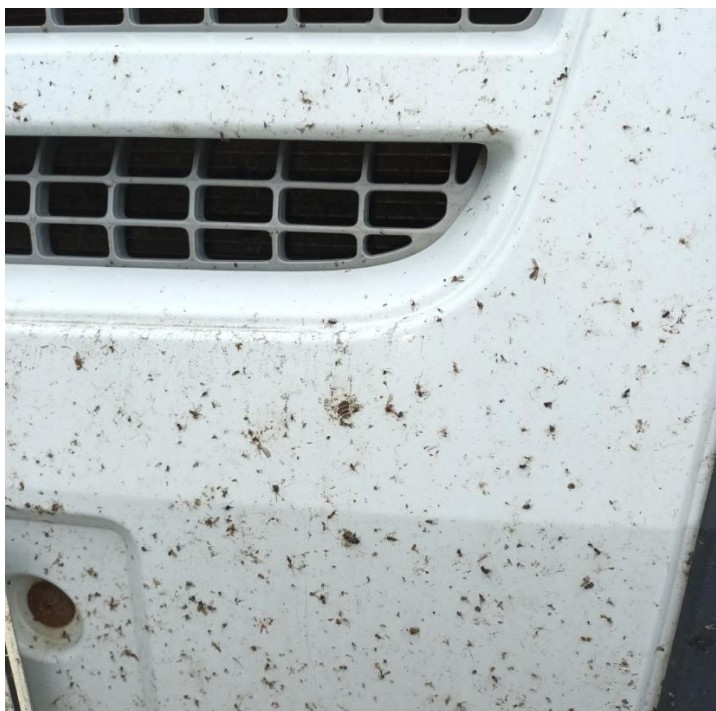

**Figure 1.** Insects (at least Culicidae, Mycetophilidae and Simulidae; identification by Dr. Jukka Salmela) smashed on the front panel of a car in the city of Rovaniemi, Finland (6 September 2022; Photo J. Jokimäki).

The use of this car-based insect food resource is inadequately studied, even if the first observation of the use of insects smashedon cars by the House Sparrow (*Passer domesticus*) in

London (U.K) was reported in 1928 [24]. After that first observation, anecdotal observations of this behavior have been reported worldwide [24–43]. However, to our best knowledge, there are no studies where the use of insects on front panels of cars by birds has been studied more widely and in a systematic way.

The main study aims of our research were to analyze (1) which bird species are able to use this car-based food resource, (2) when different species started to use this resource, (3) how common and widespread this behavior is, (4) in which habitat this behavior occurs, and (5) during what seasons birds use this food sources in Finland. To our best knowledge, no earlier observations on this behavior have been reported in Finland. We hypothesized that innovativeness, preferred habitat type, diet and migratory status, among other factors, might determine if species are using insects smashed on cars. As some bird species are more innovative than others [44], we predicted that innovative species use this food resource more often than less innovative species. Foraging innovation occurs when animals exploit novel food sources (such as insects smashed on cars) or invent new foraging techniques (picking insects from the front panel of the cars). An earlier study reported there might be a positive relationship between innovation rate and habitat generalism, but not necessarily between the innovation rate and diet breadth of birds [45]. Additionally, we predicted that urban exploiters, omnivores, and sedentary species able to use feeding sites would be able to use this food source more often than other species, partly because they might have been early-adapted to use diverse types of food stuffs occurring in urbanized areas. As there are more cars in urban than in more natural areas, we predicted that this behavior occurs more often in urban than in more natural areas. As insect food is especially important for the growth of birds, we predicted that this behavior occurs most often during the nestling and fledgling periods of the birds. Moreover, it is possible that such novel behaviors have the potential to become common and more important over time.

## 2. Materials and Methods

### 2.1. Study Area

The study was conducted on two scales, firstly on a national level in Finland. Finland has about 5,6 million inhabitants living on an area of 338,455 km$^2$. Most of the country belongs to the subarctic Köppen climate zone and to the boreal forest vegetation zone [46]. Finland has a total of 309 municipalities, most of them with less than 6000 inhabitants [47,48]. The 107 cities in Finland have about 4,4 million inhabitants; the largest one, the capital of the country, Helsinki, has 660,821 inhabitants [45]. The mean human population density in different cities varies between 1.4 (Kaskinen) and 3085 (Helsinki) inh./km$^2$ [48]. Secondly, a more detailed study was conducted at a local level in the main urban areas of the city of Rovaniemi (66° N; 25° E; 64,022 inhabitants) located in the subarctic climate zone and middle boreal vegetation zone in Finland [49]. Seasonality is typical in Finland [43]. In summer, when the insect picking period was predicted to occur, the mean daily temperature is consistently above 10 °C. Summer (a mean daily temperature > 10 °C) starts in late May in southern Finland and lasts until mid-September [50]. In Rovaniemi, located in northern Finland, summer is about two months shorter than in southern Finland. The midnight sun period (i.e., when the sun does not set) is about two and half months long in northern Finland [50]. In Finland, about 235 bird species, from which about 100 belong to Passeriformes (the main target of this paper), breed annually [51]. The egg laying period starts, for most species, during May, and the whole breeding period mostly covers May–July [51].

### 2.2. Study Methods

We used multiple methods to collect data at the two scales, the nation-wide scale in Finland and the local level scale in a single Finnish city, Rovaniemi. Data were collected by five different methods from Finland. Firstly, a questionnaire was sent to the Finnish birders via the BirdLife Finland Birdnet emailing list [49] on 26 July 2022, and repeated on 8 August and 16 September 2022. The questionnaire contained the following questions: (1) Have

you detected that birds collect/forage insects from the front panels of cars (Yes/no); (2) If yes, what species you have observed to do so (the following species names were given in advantage [House Sparrow, Eurasian Tree Sparrow (*Passer montanus*), Rock Dove (*Columba livia domestica*), Eurasian Jackdaw (*Corvus monedula*), Hooded Crow (*Corvus corone cornix*), Eurasian Magpie (*Pica pica*), European Blackbird (*Turdus merula*), Great Tit (*Parus major*), Blue Tit (*Cyanistes caeruleus*), European Greenfinch (*Carduelis chloris*), Other species (name it)]); (3) Have you observed this phenomenononly in urban areas, or also in rural areas; (4) Which kinds of habitats/areas you have observed this phenomenon (at car parking areas of hyper/super markets, parking sites along streets, own home garden, other type of site); (5) When you detected this behavior at the first time for the different species; (6) How common this behavior is (very common, common, rare or very rare); and name your observation municipality. The Birdnet emailing list has about 1000 receivers [52]. A total of 29 birders responded the query. Secondly, the same questionnaire was sent on 27 July 2022 for 11 professional ornithologists with long-term (several decades) experience in bird monitoring work in Finland. By the front panel of the car, we mean the entire front part of the car, including the register number plate, but excluding front windscreen and engine bonnet (Figure 1).

Thirdly, a data search was conducted on 19 September 2022 using the tiira.fi bird observation database [53], in which birders save their observations in Finland. The system was opened in 29 March 2006, and a number of 25 million observers was reached in June 2022. It is noteworthy that birders have also saved in the system their older observations. Currently, there are about 40,000 registered users in the system and they save about 2 million observations per year. About 6000 people use this system as their birding notebook, i.e., they can be considered serious birders. The search was directed at the same species listed in questionnaire, with two additional species, the Eurasian Bullfinch (*Pyrrhula pyrrhula)* and the White Wagtail (*Motacilla alba*), for which one observation was reported in the questionnaire/survey. Eurasian Bullfinch was detected earlier to pick insects from the front window of the car. The timeframe of the search was set to 1 January 1950 onwards, and restricted to the Finnish summer and early autumn season (1 June–30 September). The selection of this season was based on data of earlier observations related to this specific behavior in Finland. When performing the search, the system opens a list of observations that fulfil the search criteria. The list shows observations in lines where the following information is directly visible: species, observation date, municipality, detailed site, additional information and observer name. A total of 399,155 observations were checked, and the number of observations per species varied from 11,497 (Rock Dove) to 83,705 (White Wagtail; accessed on 19 September 2022 [50]). During the screening of the observations, we checked if the observation contained additional information, and if there was information related to the interested behavior.

Fourth, we collected our own old (since 1976) anecdotal bird observations from our field notebooks. This data considers only the city of Rovaniemi. Fifthly, we conducted special bird surveys during the summer–autumn 2022 in Rovaniemi. We surveyed birds and tried to detect if the birds used food resources on the front panels of cars in four types of habitats: (a) land-sharing urban areas, (b) land-sparing urban areas, (c) market/shop/gas station, etc. parking sites and (d) along road parking areas. The land-sharing areas comprised low-density housing with private houses, and the land-sparing areas comprised high-density housing with blocks of flats (apartment blocks) [54]. The land-sharing residential areas had several small and fragmented green areas, whereas the land-sparing residential areas had >50% of their green area in a single patch [54]. Birds were surveyed from a total of 5 shared (a) and 5 spared (b) type study plots (500 m × 500 m) during 13 July–31 July 2022 by using a single-visit method (1 h/plot). Birds in parking sites ([(c) *n* = 14; and (d) *n* = 6]) were surveyed during 4 August–16 September 2022 using about 10 min single visit, except one large hypermarket site (5.5 ha from which 1.7 ha was sampled; mean [range] of the other (*n* = 11) parking sites = 0.42 [0.12–0.95] ha), where one visit per week was conducted. The sample length of the road site parking sites varied from

60 to 200 m, with a mean of 120 m (*n* = 7). The number of cars at the parking sites was calculated from some (*n* = 9) parking sites, and it varied from 30 to 200 (mean = 75 cars). Correspondingly, car parks along both sides of roads had some tens of cars that constantly come and go (*n* = 7). During the field surveys, we walked slowly throughout the study sites and tried to detect birds near the cars. If birds were observed near the cars, we followed them using binoculars (Leica 8 × 40) for some minutes to observe if they picked insects from the front panels of cars. All surveys were conducted during midday and only during sunny, non-raining and non-windy weather conditions.

Because of the multiple data collection methods, there was a possibility of pseudoreplication (i.e., the same observation included twice in the data file). When saving individual observations in an Excel file, we detected only a few replicated observations. These replicated observations were later removed from the data file and where not used in this study.

The distribution ranges of all the studied species, except the southernly distributed Eurasian Jackdaw and the Eurasian Tree Sparrow, cover the whole of Finland [55].

## 3. Results

The total pooled data contained 102 observations from seven bird species using insects on the front panels of cars. Most of the observations were gathered via the questionnaire (43.5%), whereas 22.2% of the data came from the tiira.fi data base, 20.4% were the author's own observations and 13.9% of the observations came from expert ornithologists. Based on the questionnaire survey, 60% of responders stated that this behavior was currently either rare or very rare, whereas others stated that the behavior was quite common in Finland (*n* = 20).

### 3.1. Species Observed, Their Distribution and Numbers

Over half (64.1%) of the observations identified the House Sparrow, followed by the White Wagtail (10.7%), the Eurasian Jackdaw (8.7%), the Eurasian Tree Sparrow (4.9%), the Hooded Crow (4.9%), the Great Tit (3.9%) and the Eurasian Magpie (1.9%; *n* = 102; Figure 1; Table 1). The Eurasian Bullfinch was observed to pick insects from windscreen wipers.

Distribution of the observations of the House Sparrow were spread throughout Finland (Figure 2b), whereas the other species showed a more restricted distribution (Figure 2c–e). All Eurasian Jackdaw observations were seen in southern Finland (Figure 2d).

The first observation of the House Sparrow picking insects from the front panel of cars was made in the year 1971 in the core area of the capital of Finland, Helsinki (Table 1). Four years later (1975), the first observation was made for the White Wagtail, followed by the Hooded Crow (1997), Eurasian Jackdaw (2006), Eurasian Tree Sparrow (2011), Eurasian Magpie (2019), Great Tit (2020) and Eurasian Bullfinch (2022; Table 1). Since the year 2012, this behavior was observed almost yearly in the House Sparrow, whereas in other species, observations of foraging on the front panels of cars are still rare (Table 1).

**Table 1.** Number of foraging observations of different bird species (HS = House Sparrow; WW = White Wagtail; EJ = Eurasian Jackdaw; ETS = Eurasian Tree Sparrow; HC = Hooded Crow; GT = Great Tit; EM = Eurasian Magpie; and EBW = Eurasian Bullfinch) on the front panels of cars in Finland in 1971–2022. Note, some reported observations lack the observation year; therefore, these observations are not included in this table. * picking insects from the windscreen wipers.

| Species | 1971 | 1975 | 1984 | 1990 | 1995 | 1996 | 1997 | 2000 | 2006 | 2007 | 2011 | 2012 | 2013 | 2014 | 2016 | 2017 | 2018 | 2019 | 2020 | 2021 | 2022 |
|---|---|---|---|---|---|---|---|---|---|---|---|---|---|---|---|---|---|---|---|---|---|
| HS | 1 | | 1 | 2 | 1 | 2 | | 1 | | | | 3 | 3 | 1 | 2 | 1 | 6 | 9 | 5 | 2 | 19 |
| WW | | 2 | | | | | | | | | 1 | | | | | 1 | | | 1 | 1 | 2 |
| EJ | | | | | | | 1 | | 1 | 1 | | 1 | | | | | | | 1 | | 3 |
| ETS | | | | | | | | | | | | 1 | 2 | | | | | | | 1 | 1 |
| HC | | | | | | 1 | | | | | | 1 | | | 1 | | | | | | 1 |
| GT | | | | | | | | | | | | | | | | | | | | | 2 |
| EM | | | | | | | | | | | | | | | | | | 1 | | | |
| EBF * | | | | | | | | | | | | | | | | | | | | | 1 |

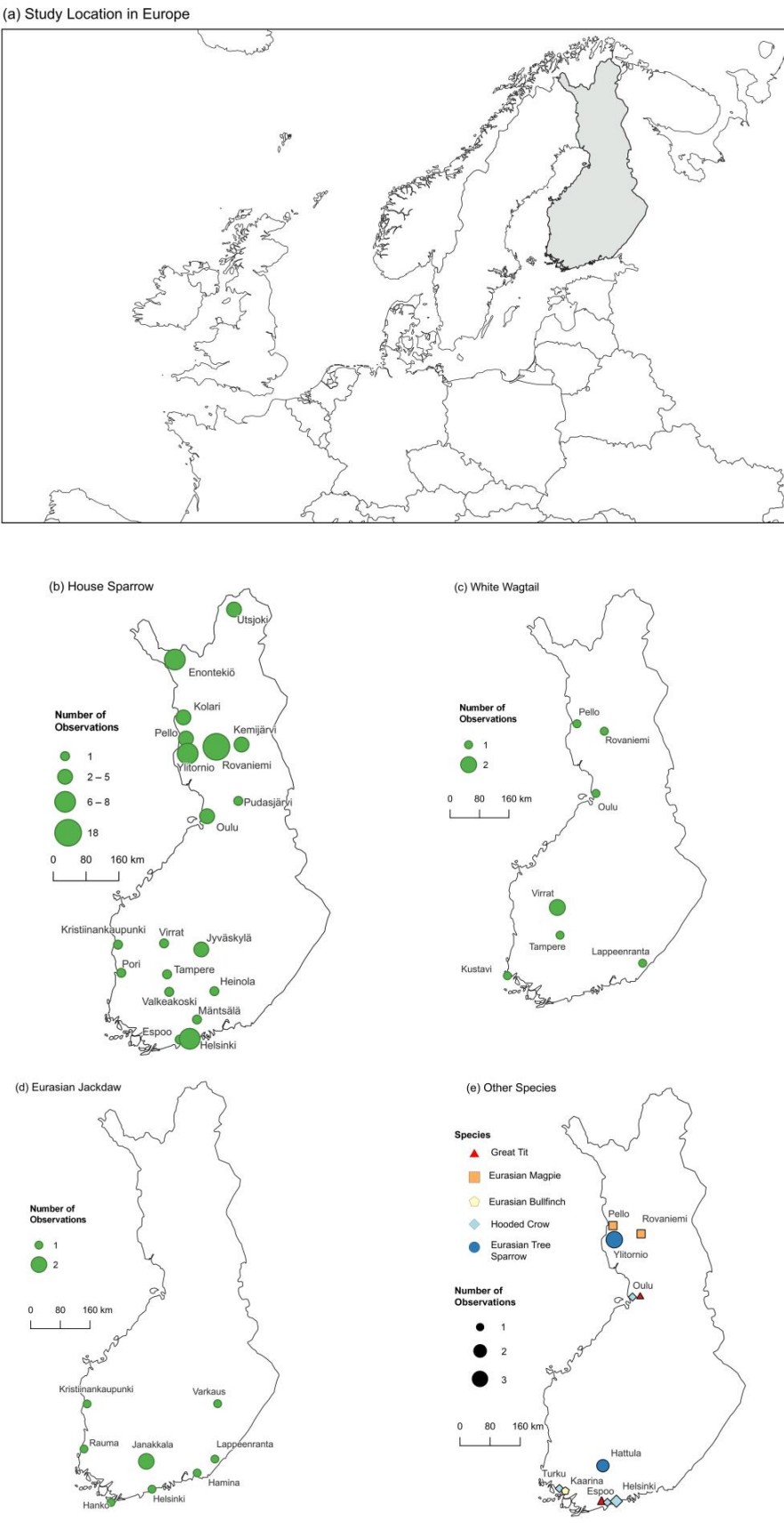

**Figure 2.** (**a**) Location of the study area; Number of observations of the (**b**) House Sparrow; (**c**) White Wagtail; (**d**) Eurasian Jackdaw; and (**e**) Other species (Great Tit; Eurasian Magpie; Hooded Crow; and

Eurasian Tree Sparrow) using insects on the front panels of cars during 1971–2022 in Finland. Municipality names of the observation sites are given. Note that the symbol scale varies between subfigures.

### 3.2. Habitats and Seasonality

Over three-fourths of the observations (78.2%) were made in urban areas, 18.8% in rural and 3.0% in suburban areas ($n = 103$). A total of 71.2% of all the House Sparrow observations were made in parking areas of hyper/supermarkets, 23.7% in parking areas within the core areas of the city, 3.4% in parking areas of gas stations and 1.7% in parking places of hamburger sites ($n = 59$). Additionally, most observations of the Eurasian Jackdaw (44.4%, $n = 9$) and White Wagtail (40.0%, $n = 5$) were conducted in hyper/supermarket areas. However, 50.0% ($n = 6$) of the Hooded Crow observations were made within the parking areas of core city areas. Due to low sample sizes, habitats of the other species are not presented here. During the author's own field surveys in the year 2022, no observations of birds foraging on the front panels of cars were made in land-sharing areas, land-sparing areas or parking sites along the roads, whereas four observations were made in parking areas of hyper/supermarkets (3 House Sparrow observations; 1 White Wagtail observation).

Most of the foraging observations were made during the mid-summer season (Table 2). The mean foraging date of the urban House Sparrow (23 July, $n = 39$) was about one week earlier than that of the rural House Sparrow (28 July, $n = 7$).

**Table 2.** Median and range dates of the foraging observations of different bird species on the front panels of cars in Finland, 1971–2022. Note that the "N" for the Eurasian Magpie and the Hooded Crow is only one.

| Species | Median Date | Date Ranges | N |
|---|---|---|---|
| House Sparrow | 17 July | 30 May–27 September | 46 |
| White Wagtail | 26 July | 2 July–8 August | 5 |
| Eurasian Magpie | | 23 July | 1 |
| Hooded Crow | | 26 July | 1 |

### 4. Discussion

Our results indicated that a total of seven bird species have been detected to use insects on the front panels of cars in Finland since 1971. However, this behavior is not yet common, since about 60% of responders stated that this behavior is currently either rare or very rare. Over half of the observations identified the House Sparrow, and about one tenth identified the White Wagtail or the Eurasian Jackdaw. Other species observed to pick insects from cars were the Eurasian Tree Sparrow, the Hooded Crow, the Great Tit and the Eurasian Magpie. The Eurasian Bullfinch was observed to pick insects from windscreen wipers. The phenomenon was distributed quite widely across the whole of Finland in the case of the House Sparrow, whereas the observations identifying the Eurasian Jackdaw were restricted only to southern Finland. Since the year 2012, insect picking behavior from the cars was observed almost yearly in the House Sparrow, whereas foraging observations of the other species are still rare. Most of the observations were conducted in urban areas and parking sites of hypermarkets. Most of the foraging observations were made during the end part of the breeding season, during July.

The richness of bird species using insects from the front panels of cars in Finland (seven species) was relatively high. When conducting a Scopus and a Google Scholar data search (accessed on September 2022) and going through *The Handbook of the birds of Europe, the Middle East and North Africa: the birds of the Western Palearctic* (1977–1994; [56–64]) and the Finnish *Pohjolan Linnut Värikuvin* handbooks (1963, 1967 [65,66]), we were able to find corresponding reports of only seven species worldwide: the House Sparrow (multiple reports), the Carrion Crow (*Corvus corone* corone) [40], the White Wagtail (*Motacilla alba alba*) [40], the Herring Gull (*Larus argentatus*) [40], the Blackbird (*Turdus merula*) [40], the Greenfinch (*Carduelis chloris*) [39], and the Boat-tailed Grackle (*Quiscalus major*) [34]. To

our best knowledge, here we report the first ever observations of the Eurasian Magpie, the Eurasian Tree Sparrow, the Great Tit and the Eurasian Jackdaw taking insects from the front panels of cars.

Two groups of species, sparrows and crows, were well presented in our samples, whereas no observations were conducted for some common urban species, such as the Rock Dove (*Columba livia*), Blackbird, Fieldfare (*Turdus pilaris*), Greenfinch or Blue Tit. One reason for this observation is that sparrows and the corvidae family (crows and jays) are very innovative species, compared with the other species listed above [44]. In many studies, the feeding innovation frequency has been observed to correlate positively with the relative brain size of birds [66,67]. For example, corvids have large relative brain sizes [44]. Based on our results, as well as the earlier reports, almost all observations of birds using insects on the front panels of cars identified the House Sparrow. This was not a surprise since the House Sparrow is a highly innovative species [44], urban exploiter, omnivore and sedentary [68]; all features linked for successful living in urban settings. In addition, a large number of records from this species might also reflect the fact that their density is higher in urban areas than that of other study species. However, only using the absolute number of records might be partly misleading.

Only one species, the White Wagtail, was migratory and purely an insectivore, others being sedentary or partly migratory and having an omnivorous feeding behavior. It has been observed that omnivore (generalist) species have greater food types and technical innovation rates than diet specialists, as well as larger brains, suggesting that cognitive skills benefit generalist species to expand their diet for new resources, such as insects on front panels of cars [69]. Consistent with these ideas, Sol et al. (2005, [70]) have shown that brain size indicating innovative skills may reflect the ability of species' success in colonizing novel environments, such as urban areas [71]. Additionally, in our case, all species using insects on the front panels of cars, except the White Wagtail, use commonly winter-feeding sites in Finland [72]. Moreover, all of these species are typical and abundant birds in urban areas of Finland and also in Rovaniemi [73–75].

Related to the three most observed species using insects on the front panels of cars, observations of the House Sparrow and the White Wagtail were distributed quite widely across Finland, whereas the observations of the southernly distributed Eurasian Jackdaw were made only in southern Finland. Additionally, the lower densities of the Eurasian Jackdaw up north can explain absence of observations from the northern latitudes. In addition, several participants reported that Eurasian Jackdaws picked insects from the front-sites of long-distance trains arriving at the Helsinki railway station, in southern Finland. The first observation of birds picking insects from cars in Finland (Helsinki, House Sparrow, 1971) was made about 40 years later than the first observation in London. This time difference might be due the fact that motor traffic took over gradually starting in the mid-1950s in Finland [76]. Where there were only some tens of thousands of cars in Finland during the 1940s and 1950s, the numbers increased to hundreds of thousands between the 1960s and 1970s [76]. The Finnish automobile stock continued to increase rapidly and reached the one million mark no earlier than in 1976, and since 2006, there are about three million cars in Finland [76]. Increasing numbers of cars means that more insects are smashed on cars [22,23], and are available for the use of birds. The importance of the number of cars is also highlighted in the habitats where the behavior occurs. Most observations were made in urban areas, and within urban areas, in places where many cars are present, i.e., in large-sized parking areas where cars are parked often for several tens of minutes, if not hours. Interestingly, no observations were made in residential areas of Rovaniemi city, despite intensive field surveys.

Most of the observations were made during the late phase of the breeding season, indicating that this behavior might be related to the lack of or need for nutrition-rich food for nestlings or fledglings in urbanized areas [10–14]. Indeed, we made some observations when adult House Sparrows collected insects from cars in their beak, and thereafter clearly flew to their nesting hole. Additionally, we detected once that an adult House Sparrow

was taking insects from cars and then feeding their three fledglings near the car. This observation raised the question of whether fledglings are learning this behavior from their parents in this way. However, more data are needed to support this conclusion.

As with all ecological and questionnaire studies, our study also has some weaknesses. Several responders indicated that they either do not remember if they have or have not detected this behavior, and if detected, they did not even remember the observation year or the species. Many responders also informed us that they have not paid any attention to this behavior. Therefore, it might be possible that this behavior is more common than reported in this study, and the given species-specific first detection years might be too late. Additionally, it is possible that people who have not witnessed this behavior might not have completed the survey, even if we also asked people to report negative observations. In this case, the prevalence of this behavior would be overestimated. In addition, the number of responders to our questionnaire was not very large.

We used several methods to study the phenomenon. By using a questionnaire type of research method, one can easily acquire a broad overview of the phenomenon. However, field studies are needed to acquire more comparable and quantitative data. In these field studies, availability of resources (number of cars with and without insects) as well as bird species and their numbers at the study sites should be estimated. As this phenomenon seems to be most popular in parking areas with plenty of cars that stay at the site for a long time, e.g., parking areas of hypermarkets, large-sized parking outdoor parking sites might be the best research sites to study this phenomenon in more detail.

Several interesting questions are still open for the further research. Eventually, we need quantitative measurements to determine how common this behavior really is. At the moment, what we know roughly what species this phenomenon considers, when and where it occurs. It might be interesting to study whether the relative paucity of observations outside cities is related to the smaller numbers of cars or the greater natural availability of insects. In addition, we do not know if there are sex- or age-specific differences in this behavior, although most of our House Sparrow field observations were considered adults. More importantly, it would be interesting to know if this artificial insect food source has any impact on breeding success and development of nestlings/fledglings of those species and individuals that use that food source. Related to the food source itself, it might be interesting to know if the car type, driving speed and the route of the car influences the number of insects smashed on the front panels and windows of cars (see, e.g., [22]). Another consideration is whether this behavior requires a certain level of tolerance of human presence, as it occurs in crowded areas. This might exclude species that initiate flight at a large distance. Moreover, with more data on this phenomenon, it will be possible to determine whether this habit is more prevalent in more generalist species and those with a relatively larger brain.

## 5. Conclusions

Despite the fact that the Editorial note of *The British Birds* (Vol. 95, page 652) already stated in 2002 that "This habit of feeding on dead insects stuck to trains and motor vehicles seems to be fairly widespread and is now well documented", our article indicates that this behavior is not so widespread, and it is not a habit for many species, at least in northern countries such as Finland. However, several new species were detected relatively recently to use this artificial food resource. This gives potential for such behavior to become more common, e.g., in crows. We will encourage researchers from other countries to conduct the same kind of questionnaire and field studies to understand in more detail and more widely where this behavior occurs, which species exhibits it and what benefits individuals using this behavior achieve from these activities.

**Author Contributions:** Conceptualization, J.J.; methodology, J.J.; formal analysis, J.J.; investigation, J.J. and M.-L.K.-J.; writing—original draft preparation, J.J.; writing—review and editing, J.J. and M.-L.K.-J. All authors have read and agreed to the published version of the manuscript.

**Funding:** This research received no external funding.

**Institutional Review Board Statement:** Not applicable.

**Data Availability Statement:** The data presented in this study are available on request from the corresponding author. The data are not publicly available due to the reason that the data file contain personal information details.

**Acknowledgments:** We thank all the participants providing their observations for our use. Additionally, we thank the regional ornithological bird societies (Lapin LTY; Kanta-Hämeen LTY; Pohjois-Pohjanmaan LTY; Päijät-Hämeen LTY and Suomenselän LTY) for allowing us to use their observations saved in the tiira.fi bird observation data base. Special thanks to the 11 Finnish professional ornithologists sharing their views related to the phenomenon studied. We thank Jukka Salmela, from the Regional Museum of Lapland, Finland, for the identification of the insects in Figure 1 and discussion about the abundance of insects in Lapland, northern Finland. Lastly, we thank system analyst Arto Vitikka from the Arctic Centre for producing the maps.

**Conflicts of Interest:** The authors declare no conflict of interest.

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
