# Peer review of "Urban Birds Using Insects on Front Panels of Cars"

_2673-6004, doi:10.3390/birds4010002_

Round 1
Reviewer 1 Report
This is an interesting contribution of a northern perspective to the question to what extent insects smashed on cars can be significant food supply for urban birds. It is interesting that the authors show a much less pronounced phenomenon in contrast to other countries.
In my opinion the paper is of appropriate length and depth and can be accepted. Just a few minor things to consider:
line 29 "The first observation was made about the House Sparrow on 1971,": a few lines below you describe the first records were from 1929, please clarify.
line 66: the statement that insect declines in urban areas are due to poorer air quality is a bit too simple. There are other factors as well, like habitat loss, loss of host plants, intensive insecticide use in private gardens etc.
line 100: isn't it a bit trivial to state that the usage of insects on cars is more often found in places with more cars? I guess this rather shall aim to the question where it is more likely to develop, but then the sentence needs to be rephrased.
line 101: I doubt insect food is only important for body growth but rather for the whole development / ontogenesis.
line 198: it would be good to provide the reade with the info here where the northern distribution limit of Jackdaws in Finland is. Could that be the reason why there are no instances observed in the north?
The language is, on one hand, mostly fluid, but on the other hand there are numerous real errors in grammar or simply wrong words are used. Here is a list of the most obvious ones, much likely not complete:
line 27: "The phenomenon was distribution quite widely": ... was distributed...
line 28: "Jackdaws which observations": .... Jackdaws with observstions...
line 38: "to conduct similar kinds studies": ...similar kinds of studies...
line 68: "in the highly urbanizes city areas.": urbanized
line 81: "there is no studies": are no studies
line 84: "different species has started": have started
line 85: "in which habitat this behavior occur": occurs
line 108, 121: "belongs on... and on...": belongs to
line 111:"density in different cities very between": vary between
line 117: "a man daily temperature > 10°": I hope that very much even when I am convinced Fins are pretty cool ;-) - I guess this should read "mean daily temperature"
line 128: "was send": sent
line 132: "species names was given in advantage": speceies names were given in advance
line 153: "Pyrryhyla": Pyrrhula
line 155: "was sent 1.1.1950": I don't think so. Should it read "was set to 1.1.1950"?
line 169/170: "The land-sharing areas constitutes from": constitute of
line 171: "with block-of-flats ()." There is something missing in the brackets. "block-of-flats" should be written "block of flats" but I suggest the more common term "appartment blocks".
line 220: "of the all House Sparrow observations": delete "the"
line 249: "was distribution quite widely": distributed
line 265: "able to found": find
line 289: "have showed that" shown
line 304: "he numbers crowed to about hundreds of thousands": not fully sure here. Should that mean "growed"? Then the correct form would be "grew". Maybe "increased" would be more appropriate here?
317: "and thereafter flow obviously in their nesting holes.": flew
Author Response
This is an interesting contribution of a northern perspective to the question to what extent insects smashed on cars can be significant food supply for urban birds. It is interesting that the authors show a much less pronouned phenomenon in contrast to other countries.
In my opinion the paper is of appropriate length and depth and can be accepted.
OUR RESPONSE: Thank you for your nice words towards our ms.
Just a few minor things to consider:
line 29 "The first observation was made about the House Sparrow on 1971,": a few lines below you describe the first records were from 1929, please clarify.
OUR RESPONSE: The year 1971 refers to our own result from Finland, whereas the year 1929 on L 76 refers London; we edited L76 as London (U.K.) for the clarity.
line 66: the statement that insect declines in urban areas are due to poorer air quality is a bit too simple. There are other factors as well, like habitat loss, loss of host plants, intensive insecticide use in private gardens etc.
OUR RESPONSE: You are fully right, therefore we have edited this like following, and added two references to support this: “In addition, due to the poorer air quality, loss of suitable host plants, intensive insecticide use, habitat loss and fragmentation the insect abundance in urban areas can seldom be as high as in more natural areas
Fenoglio, M. S.; Rossetti, M. R.; Videla, M. Negative effects of urbanization on terrestrial arthropod communities: A meta‐analysis. Global Ecology and Biogeography, 2020, 29(8), 1412-1429.
Fenoglio, M. S.; Calviño, A.; González, E.; Salvo, A.; Videla, M. Urbanisation drivers and underlying mechanisms of terrestrial insect diversity loss in cities. Ecological Entomology, 2021, 46(4), 757-771.
line 100: isn't it a bit trivial to state that the usage of insects on cars is more often found in places with more cars? I guess this rather shall aim to the question where it is more likely to develop, but then the sentence needs to be rephrased.
OUR RESPONSE: We have deleted the sentence “Correspondingly, this phenomenon will occur more often in sites with more cars within cities (like parking areas).” away.
line 101: I doubt insect food is only important for body growth but rather for the whole development / ontogenesis.
OUR RESPONSE: you are right, we added this in the sentence, “As insect food is especially important for the development (ontogenesis) and growth of birds.
line 198: it would be good to provide the reade with the info here where the northern distribution limit of Jackdaws in Finland is. Could that be the reason why there are no instances observed in the north?
OUR RESPONSE: We have now given the distribution ranges of different study species at the end of the Materials and Methods section with a citation: “ The distribution ranges of the all studied species, except the southernly distributed Eurasian Jackdaw and the Eurasian Tree Sparrow, cover the whole Finland [52].
Yes, you are right, lack of foraging observations of the Eurasian Jackdaw from northern Finland can be related to its´ southern distribution, this is now indicated in the Discussion section: “Related to the three most observed species using insects on the front panels of cars, observations of the House Sparrow and the White Wagtail were distributed quite widely across Finland, whereas the observations of the southernly distributed Eurasian Jackdaw were done only in southern Finland. Also, the lower densities of the Eurasian Jackdaw up north can explain absence of observations from the northern latitudes.”
The language is, on one hand, mostly fluid, but on the other hand there are numerous real errors in grammar or simply wrong words are used. Here is a list of the most obvious ones, much likely not complete:
OUR RESPONSE: Thank you for your detailed reading, and pinpointing the language and misprint errors. We have corrected all of them (not shown separately below). In addition, we have paid some extra attention to our English language usage. Note also that the final version, before the proof-reading, will be checked by the MPDI language center.
line 117: "a man daily temperature > 10°": I hope that very much even when I am convinced Fins are pretty cool ;-) - I guess this should read "mean daily temperature"
OUR RESPONSE; corrected, but indeed, we Finns are very cool?
line 169/170: "The land-sharing areas constitutes from": constitute of..
line 171: "with block-of-flats ()." There is something missing in the brackets. "block-of-flats" should be written "block of flats" but I suggest the more common term "appartment blocks".
OUR RESPONSE: We have added a missing references and given the difference description of these two types. . Ibáñez-Álamo, J. D.; Morelli, F.; Benedetti, Y.; Rubio, E.; Jokimäki, J.,;Pérez-Contreras, T.; ... & Díaz, M. Biodiversity within the city: Effects of land sharing and land sparing urban development on avian diversity. Science of the Total Environment, 2020, 707, 135477.
OUR RESPONSE: we have corrected as “with block of flats (apartment blocks)
Reviewer 2 Report
This paper attempts to review the literature and existing databases to determine the incidence of birds feeding on insects smashed on cars. It highlights the general decline in insects in the introduction and the recent rise in the instances of feeding of smashed insects by certain birds groups. The paper looks into a large number of entries mostly by citizens and also by some experts. This is a very descriptive survey of the phenomenon and the methods are described with some missing details. For example how did the authors deal with entries involving psuedoreplication (of the same birds feeding repeatedly in the same parking lot), etc? However, this could be simply stated somewhere in the methods as a clarification. The data is descriptive and the conclusion based on the data is still sound. THe paper needs thorough editing as it has many stylistic and grammatical errors. The word questionary should be replaced with Questionnaire (among other things). The discussion should be shortened more as the core findings are brief. I can suggest acceptance after the changes are made.
Author Response
This paper attempts to review the literature and existing databases to determine the incidence of birds feeding on insects smashed on cars. It highlights the general decline in insects in the introduction and the recent rise in the instances of feeding of smashed insects by certain birds groups. The paper looks into a large number of entries mostly by citizens and also by some experts. This is a very descriptive survey of the phenomenon and the methods are described with some missing details. For example how did the authors deal with entries involving psuedoreplication (of the same birds feeding repeatedly in the same parking lot), etc? However, this could be simply stated somewhere in the methods as a clarification. The data is descriptive and the conclusion based on the data is still sound. THe paper needs thorough editing as it has many stylistic and grammatical errors. The word questionary should be replaced with Questionnaire (among other things). The discussion should be shortened more as the core findings are brief. I can suggest acceptance after the changes are made.
OUR RESPONSE: Thank you for your positive feelings towards our manuscript. We have clarified the description of the methods related to how we take in the account the possible pseudoreplications. We added the following text at the end part of the Material and Method section: “Because of the multiple data collection methods, there was a possibility for the pseudoreplication (i.e. the same observation included twice in the data file). When saving individual observations in the Excel file, we detected only a few replicated observations. These replicated observations were later removed from the data file.
We have also corrected all stylistic and grammatical errors, also R1 pointed us quite many misprints that are now corrected. For example, “questionary” is now replaced with “questionnaire”. Note also that the final version, before the proof-reading, will be checked by the MPDI language center. We have tried to shorten the Discussion section of the manuscript by deleting some parts, but at the same time we were forced to enlarge it due to the requests of the other reviewer.
Reviewer 3 Report
This paper examines the occurrence of a behaviour that has not attracted a lot of attention, namely, the habit of picking up smashed insects from cars. Using various approaches including surveys and field sampling, the authors were able to determine that this behaviour has occurred in Finland for quite some time and involves at least seven species, especially the House sparrow. With more data on this phenomenon, it will be possible to determine whether it is more prevalent in more generalist species and those with a relatively larger brain.
Line 86: It may be worth mentioning whether this behaviour has been observed in Finland before and also why it is interesting to study this behaviour in Finland in particular. For instance, is the decline in insect abundance steeper in Finland than elsewhere?
Line 131: Please define the parts that constitute the front panel of cars.
Line 141: Did you specify what it means for the behaviour to be common or rare (e.g. number of observations of per day or per week)? People might view the scale for this variable differently.
Line 165: What are these observations and how were they collected? I suppose it was not on the tiira.fi platform.
Line 166: Perhaps remind the readers about where this sampling occurred. What sort of sampling took place? For instance, did you carry focal observations on particular birds or noted all occurrences of the behaviour for all individuals in the field of view?
Line 168: Please define land-sharing and land-sparing areas.
Line 182: I think it would be easier to follow the results if they were broken down according to the sources of information as presented in the methods section (with numbers). All these different results could then be summarized.
Line 193: What is this sample size of 103 given that in total there were 102 observations?
Table 1: Please provide rows and column totals.
Line 261: Is species richness really high when the same number of species were recorded in other areas of the world?
Line 281: Yes perhaps house sparrows are more innovative but a large number of records from this species might also reflect the fact that their density is higher. Just using the absolute number of records might be misleading.
Line 297: Is the jackdaw present in northern Finland? Perhaps lower densities up north can explain this.
Discussion: Also people who have not witnessed this behaviour might not have completed the survey. In this case, the prevalence of this behaviour would be overestimated. The small proportion of respondents is a concern. Since you have used different approaches to evaluate the occurrence of this behaviour, it might be a good place to suggest what seems to be the best approach for future studies. Eventually, we need quantitative measurements to determine how common this behaviour really is. At the moment, what we know if when and where it occurs. Readers might want to know the best way to estimate the prevalence of this behaviour so that results can be compared across regions and time periods. It might also be interesting to wonder whether the relative paucity of observations outside cities is related to the smaller numbers of cars (again this comes back to a proper quantitative assessment of prevalence) or the greater availability of insects. Another consideration is that this behaviour requires a certain level of tolerance of human presence as it occurs in crowded areas. This might exclude species that initiate flight at a large distance.
Line 352: In my experience, it is required to have an assessment by a University Ethics Board for any survey.
Author Response
This paper examines the occurrence of a behaviour that has not attracted a lot of attention, namely, the habit of picking up smashed insects from cars. Using various approaches including surveys and field sampling, the authors were able to determine that this behaviour has occurred in Finland for quite some time and involves at least seven species, especially the House sparrow. With more data on this phenomenon, it will be possible to determine whether it is more prevalent in more generalist species and those with a relatively larger brain.
OUR RESPONSE: A good overview about our research. The last point was good, and we have included it at the end part of our manuscripts, thanks.
Line 86: It may be worth mentioning whether this behaviour has been observed in Finland before and also why it is interesting to study this behaviour in Finland in particular. For instance, is the decline in insect abundance steeper in Finland than elsewhere?
OUR RESPONSE: We added a sentence in this place: “To our best knowledge, no earlier observations about this behavior has been reported from Finland.” Unfortunately, we have no data/results if the decline of insects is steeper in Finland than in other countries. In general, irrespective the country, it is interesting to study this phenomenon.
Line 131: Please define the parts that constitute the front panel of cars.
OUR RESPONSE: sorry, we describe here only our questionnaire sheet, so we can not edit this place. However, the description is now given in the Study Methods section of the manuscript as following:“By the front panel of the car, we mean the all front part of the car including the register number plate, but excluding front windscreen and engine bonnet (Figure 1).” The figure 1 (photo) describes very well the topic.
Line 141: Did you specify what it means for the behaviour to be common or rare (e.g. number of observations of per day or per week)? People might view the scale for this variable differently.
OUR RESPONSE: Unfortunately, we did not specify this in our questionnaire.
Line 165: What are these observations and how were they collected? I suppose it was not on the tiira.fi platform.
OUR RESPONSE: “Fourthly, we searched our own bird observation archives since 1976.”; we clarified this as “Fourthly, we checked our own old (since 1976) bird observation note books if there were any suitable observations for our use. This data come mostly from north Finland.”
Line 166: Perhaps remind the readers about where this sampling occurred. What sort of sampling took place? For instance, did you carry focal observations on particular birds or noted all occurrences of the behaviour for all individuals in the field of view?
OUR RESPONSE: We have now clarified the study sites as well as sampling method used in field surveys. The following clarification was added in the text: “During the field surveys, we walked slowly throughout the study sites and tried to detect birds near the cars. If birds were observed near the cars, we followed them by binoculars (Leica 8 x 40) some minutes to observe if they picked insects from the front panels of cars.”
Line 168: Please define land-sharing and land-sparing areas.
OUR RESPONSE: We have added the missing references and given the difference description of these two types. . Ibáñez-Álamo, J. D.; Morelli, F.; Benedetti, Y.; Rubio, E.; Jokimäki, J.,;Pérez-Contreras, T.; ... & Díaz, M. Biodiversity within the city: Effects of land sharing and land sparing urban development on avian diversity. Science of the Total Environment, 2020, 707, 135477.
Line 182: I think it would be easier to follow the results if they were broken down according to the sources of information as presented in the methods section (with numbers). All these different results could then be summarized.
OUR RESPONSE: Thank you for this suggestion, but we do not want to change the order because we think that it works quite well in the present form, and neither the other two reviewers nor the Associate Editor asked such changes.
Line 193: What is this sample size of 103 given that in total there were 102 observations?
OUR RESPONSE: Sorry, a misprint, corrected 102.
Table 1: Please provide rows and column totals.
OUR RESPOSE: In our mind, the table is already too wide, and the main message (which species has most observations; how the numbers of observations have increased during the years) can be easily observed from current version of the Table even without adding row and column sums. Therefore, we did not add new columns or rows.
Line 261: Is species richness really high when the same number of species were recorded in other areas of the world?
OUR RESPONSE: In our mind it is high when we compare it for the results that have been found so far globally. Also, as Finland is located in the north, the overall bird species richness in here quite low as compared to more southern areas. Also, if one will find the same number of species from a quite small areas than has earlier found from a global level, one can state that number of species is at least relatively high. We added the word “relatively” (high) in this sentence.
Line 281: Yes perhaps house sparrows are more innovative but a large number of records from this species might also reflect the fact that their density is higher. Just using the absolute number of records might be misleading.
OUR RESPONSE: You might be right, we added a new sentence here: “In addition, a large number of records from this species might also reflect the fact that their density is higher than other study species. Just using the absolute number of records might be partly misleading.”
Line 297: Is the jackdaw present in northern Finland? Perhaps lower densities up north can explain this.
OUR RESPONSE: Yes, you are right. This species in more southernly distributed than the other species, and this was already stated in the text: “whereas the observations of the southernly distributed Eurasian Jackdaw were done only in southern Finland. However, we added a new sentence “Also, the lower densities of the Eurasian Jackdaw up north can explain absence of observations from the northern latitudes.”
Discussion: Also people who have not witnessed this behaviour might not have completed the survey. In this case, the prevalence of this behaviour would be overestimated. The small proportion of respondents is a concern.
OUR RESPONSE: You are fully right, that might be the case even if we asked people to report absence of observations. We have now highlighted this possibility at the end of the Discussion section where we mentioned weaknesses of our study. The following sentence was added “Also, it is possible that people who have not witnessed this behavior might not have completed the survey even if we asked people to report negative observations. In this case, the prevalence of this behavior would be overestimated. In addition, the number of responders for our questionnaire is not very large.”
Since you have used different approaches to evaluate the occurrence of this behaviour, it might be a good place to suggest what seems to be the best approach for future studies. Eventually, we need quantitative measurements to determine how common this behaviour really is. At the moment, what we know if when and where it occurs. Readers might want to know the best way to estimate the prevalence of this behaviour so that results can be compared across regions and time periods.
OUR RESPONSE: A good idea, thanks. We give now some suggestions at the end part of the Discussion. The following sentence were added at the end of the Discussion:
“Eventually, we need quantitative measurements to determine how common this behavior really is. At the moment, what we know if, when and where it occurs.”
“We used several methods to study the phenomenon. By using a questionnaire type of research method, one can easily get a broad overview of the phenomenon. However, field studies are needed to get more comparable and quantitative data. In these field studies, availability of resources (number of cars with and without insects) as well as bird species and their numbers at the study sites should be estimated. As this phenomenon seems to be most popular in the parking areas with plenty of cars that stay at the site a long time, e.g. parking areas of hypermarkets might be the best research sites to study this phenomenon more detailed.”
It might also be interesting to wonder whether the relative paucity of observations outside cities is related to the smaller numbers of cars (again this comes back to a proper quantitative assessment of prevalence) or the greater availability of insects.
OUR RESPONSE: Yes, that might be possible. However, it is also interesting that observations were conducted also in the very north, parking places of remote parts of Finland. We added this sentence in the Discussion section.
“It might also be interesting to study whether the relative paucity of observations outside cities is related to the smaller numbers of cars or the greater availability of insects.
Another consideration is that this behaviour requires a certain level of tolerance of human presence as it occurs in crowded areas. This might exclude species that initiate flight at a large distance.
OUR RESEPONSE: Good point, according to our own urban/rural FID studies, it might be true that this phenomenon might be more common in species with short FIDS. We added this idea in the Discussion: “Another consideration is that if this behavior requires a certain level of tolerance of human presence as it occurs in crowded areas. This might exclude species that initiate flight at a large distance.”
Line 352: In my experience, it is required to have an assessment by a University Ethics Board for any survey.
OUR RESPONSE: No such statement in needed here in Finland or in our university when we are not catching, handling or doing any experiments with individuals. In this study, we are only looking for birds and their behavior.
Round 2
Reviewer 2 Report
The suggestions have been implemented.